# The Role of COX-2 and PGE2 in the Regulation of Immunomodulation and Other Functions of Mesenchymal Stromal Cells

**DOI:** 10.3390/biomedicines11020445

**Published:** 2023-02-03

**Authors:** Agnieszka Kulesza, Leszek Paczek, Anna Burdzinska

**Affiliations:** 1Department of Immunology, Transplantology and Internal Diseases, Medical University of Warsaw, 02-006 Warsaw, Poland; 2Department of Bioinformatics, Institute of Biochemistry and Biophysics, Polish Academy of Sciences, 02-106 Warsaw, Poland; 3Department of Physiological Sciences, Institute of Veterinary Medicine, Warsaw University of Life Sciences, 02-776 Warsaw, Poland

**Keywords:** prostaglandin E2, cyclooxygenase 2, mesenchymal stromal cells, immunomodulatory properties, cell therapy, proliferation, migration, differentiation

## Abstract

The ability of MSCs to modulate the inflammatory environment is well recognized, but understanding the molecular mechanisms responsible for these properties is still far from complete. Prostaglandin E2 (PGE2), a product of the cyclooxygenase 2 (COX-2) pathway, is indicated as one of the key mediators in the immunomodulatory effect of MSCs. Due to the pleiotropic effect of this molecule, determining its role in particular intercellular interactions and aspects of cell functioning is very difficult. In this article, the authors attempt to summarize the previous observations regarding the role of PGE2 and COX-2 in the immunomodulatory properties and other vital functions of MSCs. So far, the most consistent results relate to the inhibitory effect of MSC-derived PGE2 on the early maturation of dendritic cells, suppressive effect on the proliferation of activated lymphocytes, and stimulatory effect on the differentiation of macrophages into M2 phenotype. Additionally, COX-2/PGE2 plays an important role in maintaining the basic life functions of MSCs, such as the ability to proliferate, migrate and differentiate, and it also positively affects the formation of niches that are conducive to both hematopoiesis and carcinogenesis.

## 1. Introduction

The synthesis of prostaglandins (PG), including PGE2, can occur in any cell of the mammalian organism [1], and the primary substrate for their production is arachidonic acid (AA)—a component of cell membrane phospholipids. Under the influence of various stimuli such as cytokines, mitogens, hormones or factors released from damaged cells, the activation of cytosolic phospholipase A2 (cPLA2) takes place, which in turn leads to the translocation of membrane AA into the cell interior. The intensity of this reaction determines the final amount of produced eicosanoids, i.e., bioactive molecules originating from arachidonic acid or other polyunsaturated fatty acids [2]. Then, at the luminal side of endoplasmic reticulum (ER)-membranes, AA is immediately oxygenated to the Prostaglandin H2 (PGH2). This process is mediated by the Prostaglandin endoperoxide synthase (PTGS). This enzyme has two isoforms and several alias names: prostaglandin G/H synthase (PGHS), or the most commonly used cyclooxygenase (COX). Prostaglandin H2 is preferentially converted into prostaglandin E2 in the reaction catalyzed by three types of PGE2 synthases (PGES): cytosolic (cPGES) and microsomal prostaglandin E synthases 1 and 2 (mPGES-1/2) [3,4,5]. PGE2 acts by binding to one of four G protein-coupled receptor (GPCR) subtypes: EP1, EP2, EP3, or EP4. These receptors have different localization within the cell, and also show differences in signal transduction and molecular effect induced, which makes PGE2 a greatly versatile molecule [6]. The schematic PGE2 biosynthesis process is presented in Figure 1.

Both COX isoforms COX-1 and COX-2 have been documented to contribute to the production of prostaglandins, but their roles differ in details. COX-1 is responsible for the constitutive production of prostaglandins, which are involved in the maintaining the body’s homeostasis [7]. In turn, the synthesis of PGs catalyzed by COX-2 is induced by inflammatory environment [8]. The latter is related to the appearance of inflammatory signs, such as pain and fever [9,10]. The production of PGE2 during inflammation is mainly mediated by COX-2 pathway [11]. Nevertheless, it has also been shown that this process can also occur with the participation of COX-1 and cytosolic prostaglandin E2 synthase (cPGES), e.g., in the stomach [12], or the central nervous system (CNS) [13].

Inhibition of PGE2 production has been an essential anti-inflammatory strategy since 1897 when Felix Hoffman (Bayer industry) synthesized pure acetylsalicylic acid which was later sold under the trade name of aspirin [14]. Currently, non-steroidal anti-inflammatory drugs (NSAIDs) are one of the most popular medications in the world, accounting for approximately 5% of all prescriptions [15]. NSAIDs are used extensively as antipyretics and analgesics in treating various types of pain. Their mechanism of action is based on cyclooxygenase enzymes inhibition. Although the vast majority of NSAIDs have the variable ability to inhibit both COX isoforms at therapeutic concentrations, the selectivity towards COX-1 and COX-2 is now the basis of their classification systems. One of these systems divides COX inhibitors into four main categories: non-selective, complete inhibitors of both COX-1 and COX-2 (indomethacin, acetylsalicylic acid, diclofenac, naproxen, ibuprofen), complete inhibitors of COX-1 and COX-2, although with a specific preference for COX-2 (meloxicam, celecoxib, nimesulide, etodolac), potent inhibitors of COX-2, although with weak inhibitory action against COX-1 (NS-398) and weak inhibitors of both COX-1 and COX-2 (sulfasalazine, sodium salicylate, nabumetone) [16]. NSAIDs also differ in the degree of reversibility of COX inhibition, from fully reversible like ibuprofen to completely irreversible like aspirin [17].

In recent years, cell transplants have become one of the new therapeutic methods aimed at the modulation of inflammatory processes. Mesenchymal stromal cells (MSCs) are widely studied in this field. MSCs are currently defined as cells displaying fibroblastic morphology, able to proliferate in monolayer on plastic surfaces, capable of osteogenic, chondrogenic and adipogenic differentiation and presenting a specific set of surface antigens that includes expression of CD90, CD73, CD105 and lack of expression of hematopoietic markers [18]. Their primary niche is the bone marrow, where MSCs support hematopoietic stem cells (HSCs) proper activity by both secretory factors and direct interactions [19]. Whether MSCs can migrate from the bone marrow via circulation and become successfully recruited in other sites of organism in a substantial amount is still a matter of debate [20,21,22,23]. However, it is clear that cells meeting the MSC identification criteria can be found in other tissues and organs. A significant part of the physiological role of MSCs is to affect the behavior of other cells. They are known to suppress inflammation and induce immune tolerance by interactions with various immune cells including T cells, macrophages, dendritic cells, NK cells, B cells. There are several described mechanisms by which MSCs exert their immunomodulatory effect which generally involve both cell-to-cell contact and action through soluble agents. As described in the recent reviews [20,21], MSC immunomodulatory activity is linked to the expression/secretion of several molecules such as PD-L1 (ligand for Programmed Cell Death Protein 1), IDO-1 (Indoleamine 2,3-Dioxygenase 1), FasL (Fas ligand) iNOS (inducible Nitric Oxide Synthase), TGF-β (Transforming Growth Factor beta), HGF (Hepatocytes Growth Factor), HLA-G (Human Leukocytes Antigen G), galectin-1, IL (Interleukin)-6, IL1-RA (IL1-Receptor Antagonist), and Prostaglandin E2. PGE2 seems to be an important part of this activity. It has been even suggested that the level of PGE2 secretion by MSCs could be a marker for predicting the MSC transplantation efficacy when immunomodulatory effect is desired. Some in vivo studies suggested that the level of PGE2 secretion correlated with the in vivo immunomodulatory effect of MSC grafting [24]. It is reasonable to define exactly what mechanisms are involved in the processes related to the activity of MSC and PGE2. The purpose of this review is to present and discuss the available data on the role of COX-2 and PGE2 in the immunomodulatory effect of MSCs. In addition, since PGE2 is a pleiotropic molecule, the current knowledge on the influence of PGE2 on other vital features of MSCs will also be presented.

## 2. PGE2 as Part of Immunomodulatory Activity of MSCs

The ability of MSCs to regulate an immune response is widely reported in the literature and is of interest for cell therapy research in immune-mediated conditions. As mentioned above, PGE2 is currently considered one of the most important immunomodulatory factors secreted by MSCs. Mesenchymal stromal cells constitutively express COX-2 and PGE2. Unstimulated human BM-MSCs accumulate PGE2 in medium already in the first hours of culture [24]. It should be remembered that MSCs are responsive cells that can change their phenotype and secretion profile depending on the surrounding signals [25]. The inflammatory environment stimulates MSCs to release numerous soluble trophic factors, such as cytokines, chemokines, and growth factors, which generally exert anti-inflammatory, tolerogenic and pro-healing effects [26]. It seems to refer also to PGE2 secretion—although MSCs secrete PGE2 constitutively, its levels vary depending on the surrounding environment. For example, it has been shown in mice that IL-6 can upregulate PGE2 production by BM-MSCs through increasing COX-2 expression [27]. Moreover, murine MSCs highly increased the secretion of both COX-2 and PGE2 in a pro-inflammatory environment induced in vitro by addition of interferon gamma (IFNγ) and tumor necrosis factor (TNF) [28]. In human MSCs (BM-derived), the relationship between PGE2 secretion and pro-inflammatory factors is less thoroughly explored; however, it has been reported that PGE2 production is significantly increased in the presence of pro-inflammatory macrophages [29].

### 2.1. Macrophages (Mɸ)

One of the best-documented immunomodulatory properties of MSCs is their ability to affect the differentiation of macrophages. MSCs promote macrophage polarization towards the anti-inflammatory M2 phenotype. This feature of MSCs seems to be universal among species as it has been shown in mice, rat, and human MSCs [30]. Moreover, our recent study demonstrated that human MSCs isolated from different tissues (bone marrow vs. Wharton’s jelly) affect differentiation of macrophages to a similar extent [31]. Based on the rodent studies, it is postulated that this activity is mediated by IL-6, IDO, and PGE2 [32]. In in vitro murine study, macrophages co-cultured with BM-MSCs showed a decreased synthesis of pro-inflammatory cytokines (TNF, IL-1β, and IL-6) and increased production of IL-10 in comparison to macrophages in mono-culture. This process was shown to be related to increased PGE2 secretion and mediated through iNOS and COX-2, as well as EP2 and EP4 receptors [33,34,35]. This mechanism may explain the role of macrophage polarization by MSCs in the context of wound healing. PGE2 derived from MSCs acts on the EP2 and EP4 receptors on macrophages, activating adenylate cyclase and increasing the level of cAMP. This leads to the activation of PKA and the subsequent phosphorylation of the cAMP-responsive binding member (CREB) (Serr113), an important factor in macrophages in the context of wound healing, enhancing C/EBP-β transcription. C/EBP-β promotes the expression of Arg1 and Mrc1 as well as IL-10, which inhibit the activity of M1 macrophages [36]. As mentioned above, MSCs can be found in different sites of organisms. This refers to both prenatal and postnatal tissues. An interesting source of MSCs seems to be decidua, the outermost region of the gestational membranes. Rogers et al. examined the interactions between decidual stromal cells (DSCs) and macrophages and revealed that DSC-derived prostaglandins impair the production of TNF by macrophages [37]. They additionally showed in mice with induced bacterial chorioamnionitis that macrophages accumulated at sites of bacterial invasion, and the level of PGE2 in amniotic fluid was increased. The authors concluded that DCSs can regulate the response of macrophages to bacterial chorioamnionitis in a paracrine manner, through the production of PGE2 [37].

It has been postulated that MSC-derived PGE2 may affect macrophages and, consequently, therapeutic efficacy of this population also in vivo. A study in which BM-MSCs were administrated in mice with induced acute liver failure (ALF), PGE2 has been identified as an important factor regulating the therapeutic effect of MSCs in the reduction of inflammation and death of hepatocytes. PGE2 secreted by MSCs inhibited TGF-β-activated kinase 1 (TAK1) signaling and reduced the activation of NLRP3 inflammasome in liver macrophages leading to diminished level of pro-inflammatory cytokines. At the same time, this prostaglandin induced the polarization of macrophages towards the anti-inflammatory M2 profile via STAT6 and mechanistic target of rapamycin (mTOR) signaling, limiting the development of inflammation and liver damage [38]. Similar results were obtained in a study of diabetic cardiomyopathy in a rat model. The infusions of adipose tissue-derived MSCs (AT-MSCs) alleviated metabolic abnormalities and preserved the structure and function of the heart, increasing the number of M2 macrophages and reducing myocardial inflammation. Further studies confirmed the importance of PGE2 in this process, showing no effect of MSCs on macrophages after treatment with COX-2 inhibitors [39]. Data regarding the role of PGE2 in the interaction of human BM-MSCs and macrophages are much more limited [40]. One group determined that human placenta-derived MSCs administrated in the insulin growth factor 1 (IGF-1) enriched chitosan-based hydrogel ameliorated experimentally induced colitis in mice. The clinical improvement was accompanied by an increase in macrophage polarization towards M2 observed in the tissues. This, in turn, correlated with an increase in the expression of *COX1* and *COX2* genes, as well as with the level of PGE2. The authors concluded that the clinical improvement in mice was mediated by polarization of macrophages to M2 phenotype induced by MSC-derived PGE2 [41]. Another study analyzed in vitro interactions between human BM-MSCs and primary blood-derived macrophages [29]. The authors confirmed that MSCs promote polarization of macrophages into anti-inflammatory phenotype, and demonstrated that this activity was significantly stronger when cells were cultured in a hydrogel than on a standard 2D plastic surface. Moreover, they demonstrated that MSC-derived PGE2 diminished the secretion of TNF by macrophages, but had no significant effect on the production of IL-10. They additionally showed that treatment of MSCs with a macrophage-conditioned medium increased PGE2 secretion. This indicated that PGE2-related interplay between MSCs and macrophages is reciprocal and based on positive feedback. All these data suggest that the MSC-macrophage interaction is complex and PGE2, although important in this process, is among several involved factors rather than the dominant one. The overall effect of MSCs on macrophages as well as other immune cells mediated by PGE2 is shown in Figure 2. The mechanisms by which MSCs exert the effect on activity of immune cells via COX-2/PGE2 pathway are summarized in Table 1. 

### 2.2. Dendritic Cells

MSCs were shown to significantly affect the behavior of dendritic cells (DCs), a population of specialized antigen-presenting cells crucial for T cell activation and differentiation. MSCs are believed to be able to change DC phenotype, cytokine release, differentiation and maturation, as well as affect their ability to present antigen [42]. Again, PGE2 is believed to be one of key players in these interactions. It was reported that PGE2 produced by BM-MSCs is responsible for inhibiting the early maturation of DCs from peripheral blood monocytes in human cells [43]. In the presence of BM-MSCs, monocytes did not acquire the surface phenotype typical of immature (CD14−, CD1A+) or mature (CD80+, CD86+, CD83+) DCs, did not produce IL-12 and did not induce activation or proliferation of T cells [43]. Similar results were obtained in the study on human 6-sulfo LacNAc dendritic cells (SlanDCs), representing a major subpopulation of human blood dendritic cells. Human BM-MSCs impaired SlanDC maturation and their ability to secrete pro-inflammatory cytokines in a PGE2-dependent manner. In addition, SlanDCs cultured in the presence of MSCs could not stimulate the proliferation of CD4+ and CD8+ T cells and differentiation of naive CD4+ T cells into Th1 cells [44]. Another study on human cells has shown that AT-MSCs stimulated with pro-inflammatory cytokines (TNF, IL-6, and IL-1β), through PGE2, suppressed the production of DC-derived osteopontin (OPN)—a pleiotropic factor abundantly secreted in inflammatory environment and generally classified as pro-inflammatory cytokine. Interestingly, the unstimulated AT-MSCs induced the opposite effect—promoting OPN production by DCs [45]. PGE2-dependent interaction between MSCs and dendritic cells was also shown in vivo, in murine models. An administration of MSCs (of undetermined origin) in mice with induced severe acute liver failure (fulminant hepatic failure, FHF) stimulated DCs into regulatory phenotype. This process depended on phosphoinositide 3-kinase, with a significant role in PGE2 and the EP4 receptor [46]. Additionally, a study with the use of MSCs from adipose tissue in mice with induced chronic encephalitis confirmed a decrease in lipopolysaccharide (LPS)-induced DC maturation in vitro. PGE2 involvement was confirmed by adding indomethacin, the COX-1/2 inhibitor, to the culture, which reversed the DC maturation blockade [47]. To summarize this part, although the effects of PGE2 alone on dendritic cells have been shown to be complex and include both inhibitory and stimulating effects depending on the stage of development of DCs, the effect of MSCs via PGE2 on DCs appears to be unequivocally inhibitory. It is likely that this consistently inhibitory effect is due to the contribution of other immunomodulatory factors secreted by MSCs.

### 2.3. Natural Killer Cells

Natural killer (NK) cells constitute an important population of innate immunity cells showing features of natural cytotoxicity, which makes them the main line of defense in the fight against viruses and cancer. It has been shown that MSCs and NK cells undergo mutual interactions in vitro [48,49]. Activated NK cells can kill MSCs, while MSCs strongly inhibit interleukin-2 (IL-2)-induced NK cell proliferation, cytotoxic activity and cytokine production. The observed effect was related to a decrease in the expression of NKp30, NKp44 and NKG2D surface NK cell receptors and the impaired production of IFNγ. One study tested the hypothesis of whether the inhibition of NK cell proliferation and cytotoxic activity in the presence of human BM-MSCs was mediated by indoleamine 2,3-dioxygenase (IDO) and PGE2 [50]. The authors have shown that the effects of IDO and PGE2 in this interaction may be synergistic. They demonstrated that only treatment of NK cells with inhibitors of both of these compounds (NS-398-PGE2 inhibitor and 1-M-Trp- IDO inhibitor) reversed an inhibitory effect of BM-MSCs on the proliferation and cytotoxic activity of NK cells [50]. Scientists explained this phenomenon by the fact that PGE2 in an autocrine manner can stimulate de novo expression of IDO mRNA in MSCs. Thus, the induction of IDO expression in MSCs can occur in two ways: directly, by stimulation with inflammatory cytokines like IFN-γ and TNF, or indirectly, by increasing the secretion of PGE2 [51]. The inhibitory effect of PGE2 (as factor added to the culture) on IL-15 activated NK cells was also demonstrated by Joshi et al. NK cells cultured for two days in the presence of 10–200 ng/mL PGE2 significantly reduced cytotoxicity and inhibited the production of IFN-γ at the transcriptional and secretory levels. The suggested mechanism of action of PGE2 is based on the reduction in the surface expression of the gamma (c) chain of the IL-15R complex on NK cells [52]. Another issue is the suppression of NK cells activity caused by MSCs-mediated increase in PGE2 level observed in various tumors which will be discussed later in this paper [53,54,55].

### 2.4. B Cells

The immunomodulatory properties of MSCs in terms of interactions with B cells are poorly understood. The available data are sparse and not consistent. A work published in 2008 evaluating interaction of human bone marrow MSCs with B cells has shown that MSCs increase B cells viability while inhibiting their proliferation and differentiation [56]. Ji et al. studied interactions between human umbilical cords (UC)-MSCs and B cells, and came to different conclusions. They have shown that UC-MSCs were responsible for increase in B cell activity: higher proliferation rate, increased differentiation into plasma cells, and an intensified production of antibodies (IgM, IgA, IgG) in vitro. By using specific inhibitors, the authors demonstrated that these interactions were mediated by PGE2, but not IL-6. Furthermore, they confirmed these results in vivo in mice, showing a significant increase in T-cell-dependent and independent antibody production in animals treated with BM-MSCs. However, in vivo experiments did not include using inhibitors of COX-2/PGE2 pathway so the mechanism of accelerated immunoglobulins (IgM, IgG) production has not been identified in this study [57]. In turn, studying the effect of BM-MSCs administration in mice with experimentally induced allergic conjunctivitis (EAC) have shown a reduced release of IgE by B cells in a COX-2-dependent mechanism [58]. Similar conclusions were drawn from a study on a mouse model of atopic dermatitis (AD). The xenotransplantation of human AT-MSCs to mice with induced AD resulted in amelioration of clinical signs and concomitant decrease in IgE level. Further in vitro experiments conducted within the same study indicated that COX-2 pathway is essential in inhibiting human B cell maturation mediated by MSCs [59]. Both of the latter mentioned studies suggested that MSCs could have beneficial effect in the treatment of allergic diseases due to COX-2-dependent mechanisms. Two other studies evaluated the effect of MSCs on IL-10 secretion by B cells. Hermankova et al. reported that murine IFNγ-primed BM-MSCs significantly decreased IL-10 secretion by LPS-activated B cells and that this effect was partially mediated through COX-2 pathway [60]. Chen et al., in turn, demonstrated that BM-MSCs induced IL-10 production by B cells and that this activity was also mediated by COX-2/PGE2 pathway. These results seem to be inconsistent with data presented by Hermankova et al. [60]. However, Chen et al. conducted experiments on human material, and B cells were represented by CD23+CD43+ B regulatory cells—a relatively small subpopulation of B cells [61]. Recently, the same group showed that PGE2 promoted IL-10 producing B regulatory cells expansion [62]. Taken together, the current data on the interaction of MSCs with B cells and the participation of PGE2 in these processes are not sufficient to draw clear conclusions. It seems that the PGE-mediated effects of MSCs on B cells may vary across different subsets of B cells. More research is definitely needed in this field to better understand these interdependencies.

### 2.5. T Cells

The inhibitory effect of PGE2 on T lymphocytes has been shown in numerous studies [63,64,65]. The role of MSCs in inhibiting T cell activation and proliferation hasbeen demonstrated for the first time by Le Blanc et al. [66] and later confirmed many times [67,68,69]. One of the proposed mechanisms responsible for the inhibition of T cell activity by MSCs is related to the COX-1/COX-2 enzymes expression and PGE2 production [70], but it seems that the effect depends on the concentration of PGE2 in the tissue microenvironment. At high concentrations, T cell proliferation was inhibited with reduced levels of IL-2 and IL-2 receptors, thereby leading to transcriptional changes via the Janus-3-kinase signaling pathway. In turn, low PGE2 concentration caused alterations in CD4+ T cells differentiation. It led to the promotion of Th2/Th17 pathways while inhibiting Th1 by affecting the secreted cytokine profile. The production of pro-inflammatory IL-12 and IFN-γ was inhibited in favor of IL-4 and IL-5, promoting Th2, whereas blocking IL-12p17 supported Th17 differentiation [71]. In the interactions between MSCs and T cells, the level of PGE2 secretion depended on the cell-to-cell contact [72,73]. 

A study on equine cells showed that MSC populations, regardless of origin (bone marrow (BM), adipose tissue (AT), umbilical cord blood (UCB), and umbilical cord tissue (UC)), can inhibit T cell proliferation by a mechanism dependent on PGE2 but not IL-6 or NO. Moreover, inhibition of PGE2 restored not only T cell proliferation but also TNF, IFN-γ secretion, and increased IL-10 levels. Presented data also suggested that the precise mechanisms of T cells inhibition by MSCs may depend on the cell source. AT- and UC-MSCs were shown to induce apoptosis, while BM- and UCB-MSCs arrested T cells cycle [53]. 

An interesting observation was made in a study on the use of UC-MSCs in the treatment of systemic lupus erythematosus (SLE) [74]. It is known that accumulation of apoptotic cells contributes to the exposure of autoantigens to the immune system that eventually can lead to autoimmunity such as SLE. Zhang et al. indicated that UC-MSCs could engulf apoptotic cells, and that this process enhanced their ability to inhibit CD4+ T cell proliferation. Apoptotic cells activated the COX-2/PGE2 axis in UC-MSCs via the NF-κB signaling pathway. Moreover, SLE patients receiving MSCs showed decreased apoptosis of peripheral blood mononuclear cells and a significant increase in plasma PGE2 metabolites which was consistent with in vitro findings [74]. In the study by Burand et al., the ability of BM-MSCs to inhibit T lymphocytes is abolished after administration to tissue because of cells aggregation. Interestingly, adding budesonide, a topical glucocorticoid steroid, alongside spheroids partially restored MSCs’ immunomodulatory properties and acted synergistically with MSCs-produced PGE2 via EP2 and EP4 receptors [75]. Studies on the interactions of mice BM-MSCs and T cells in neoplastic processes show that tumor MSCs can inhibit proliferation and induce apoptosis in T lymphocytes more than cells derived from healthy tissues. At the same time, the role of PGE2 in regulating the immunomodulatory mechanisms in tumor-derived MSCs increases as opposed to NO-dependent regulation [76].

In addition to inhibiting T cell proliferation, prostaglandin E2 secreted by MSCs promotes the appearance of functional Foxp3+ T regulatory cells (Treg), a population crucial for maintaining immunological self-tolerance and preventing excessive immune response. English et al. showed that direct contact of human BM-MSCs and CD4+ cells is required to the MSCs’-induced Treg formation, and that this is partially driven through increased PGE2 and TGF-β1 expression [77]. These processes were mediated by EP2 and EP4 receptors, which activated the nuclear factor-kB (NF-kB) signaling pathway [78]. The ability of MSCs to increase the percentage of Tregs with involvement of a PGE2-dependent mechanism has been demonstrated several times in different species. Research on pigs showed that prostaglandin E2 accelerates BM-MSC-induced IL-10+IFN-γ+CD4+ regulatory T cells differentiation to enhance the immunosuppressive potency in transplant arteriosclerosis (TA) [79]. Co-culture of MSCs from glioblastoma multiforme patients with peripheral blood mononuclear cells (PBMCs) from healthy donors stimulated the secretion of PGE2 and TGF-β1, leading to an increase in Tregs and a decrease in the Th17 cells number [80]. Another group reported that PGE2 secreted by feline adipose tissue-derived MSCs reduced inflammation by increasing FOXP3+ Treg in an in vivo mouse model of colitis [81]. Researchers studying the role of PGE2 in regulating the expression of CD46—a protein promoting T cell activation and differentiation towards a regulatory Tr1-like phenotype characterized by secretion of high amounts of IL-10—came to different conclusions. The addition of PGE2 has been shown to strongly reduce the expression of CD46 in activated T cells. At the same time, CD46 activation correlated with EP4 receptor induction and a change in the profile of secreted cytokines [65]. An increase in Tregs proportion was noted following administration of IL-17A treated BM-MSCs to mice with ischemic reperfusion injury in the kidneys. This process was regulated by the COX-2/PGE2 pathway and was associated with greater protection against acute kidney injury [82]. Examination of MSCs in a rat model of myocardial infarction showed that PGE2 increased secretion of the chemokines CCL12 and CCL5, which stimulated the chemoattraction of T cells towards MSCs. In addition, it was reported that MSCs, in a PGE2-dependent mechanism, inhibited the proliferation of cytotoxic T cells and induced the production of regulatory T cells [83]. A single-arm clinical study in 16 patients with severe and refractory systemic lupus erythematosus (SLE) showed an increase in peripheral Treg cells and restoration of the balance between Th1 and Th2 after administration of allogeneic UC-MSCs from the umbilical cord. All patients achieved significant disease activity reductions, but due to the lack of control groups, it was difficult to assess the real contribution of MSCs to the observed effects [84]. It is postulated that MSCs may affect the differentiation of T cells into subsets other than Tregs using PGE2-dependent mechanism. For example, subjecting UC-MSCs to the endoplasmic reticulum (ER) stress using a non-competitive inhibitor of the ER Ca^2+^ ATPase has been shown to have a better inhibitory effect on rheumatoid arthritis CD4+CXCR5 +ICOS + follicular helper-like T cells by releasing PGE2, which enhanced their immunosuppressive effect. Increased secretion of PGE2 and IL-6 was observed, and CD4+CXCR5+ICOS+ T cells increased frequency after the addition of EP2 and/or EP4 antagonists [85].

The effect of MSCs on Th17 lymphocytes, which play a key role in the pathogenesis of many autoimmune diseases, is unclear. Tatara et al. showed that murine BM-MSCs can inhibit the differentiation of Th17 lymphocytes in an IDO- and PGE2-dependent manner [86]. Ghannam et al. demonstrated that PGE2 partially mediated the reduction in in vitro differentiation of naive CD4+ T cells into Th17 phenotype. Additionally, human BM-MSCs inhibited the secretion of IL-17, IL-22, IFN-γ and TNF, and promoted the production of IL-10 by fully differentiated Th17 cells [87]. On the other hand, another group showed that in vitro, the PGE2/EP4 axis promoted inflammation by enhancing Th1 differentiation and Th17 expansion mediated by interleukin-23. These results were confirmed in vivo, where blockade of PGE2 pathway by administration of the EP4 antagonist reduced the accumulation of Th1 and Th17 in the regional lymph nodes, inhibiting disease progression in mice with experimental autoimmune encephalomyelitis or contact hypersensitivity [88]. Similar results were obtained by Boniface et al., who confirmed the role of PGE2 in promoting the differentiation and pro-inflammatory function of human and murine Th17. Their results indicated the participation of not only EP4 but also EP2 receptor, as well as cyclic AMP pathways. PGE2 may act synergistically with IL-1beta and IL-23 by increasing the expression of cytokines characteristic of the Th17 phenotype (orphan receptor associated with the retinoic acid receptor (ROR) -gamma, IL-17, IL-17F, CCL20 and CCR6) in lymphocytes. PGE2 also regulates the production of IFN-γ and inhibits the production of the anti-inflammatory cytokine IL-10 in Th17 cells, mainly by the EP4 receptor. Furthermore, PGE2 is required to produce IL-17 in T cells—antigen-presenting cells co-culture [89]. In conclusion, the effect of PGE2 on Th17 cells remains unclear and the results presented above are contradictory. It should be noted that in the study by Tatar et al. and Grannham et al., the effect of PGE2 on Th17 lymphocytes is studied indirectly, through interactions with MSCs. The presented results showed the inhibitory effect of MSCs on the activation and proliferation of T lymphocytes, and PGE2 is the suggested mediating molecule in these relationships, whereas in the remaining articles, in which the results indicate that PGE2 promotes the differentiation, expansion and pro-inflammatory functions of Th17, the direct effect of PGE2 was investigated. Differences in the research model may be responsible for the resulting inconsistencies, suggesting that additional mechanisms are involved in the interaction of MSCs with Th17, reducing the effect of PGE2.

**Table 1 biomedicines-11-00445-t001:** Identified mechanisms by which MSCs exert the effect on activity of immune cells via COX-2/PGE2 pathway. The table presents only studies where the proposed mechanism was confirmed by inhibition of the COX-2/PGE2 pathway.

Target Cell Type	Identified COX2/PGE2-Dependent Mechanism of MSC’ Activity on Target Cells	Species Used	MSC Type	Type of the Study	Refs
Macrophages (Mɸ)	↑ secretion of IL-10	Mice	BM-MSCs	In vivo	[33]
	↑ secretion of IL-10; ↓ secretion of TNF, IL-6, IL-10p70	Mice	BM-MSCs	In vitro	[34]
	↓ activation of NLRP3 inflammasome in macrophages from the liver with induced acute failure	Mice	BM-MSCs	In vivo	[38]
	↑ number of M2 macrophages in injured myocardium of diabetic rats	Rats	AT-MSCs	In vivo	[39]
	↓secretion of TNF in M1 macrophages	Human	BM-MSCs	In vitro	[29,40]
Dendritic cells (DCs)	↓ differentiation of monocytes into immature DCs (CD14^neg^, CD1a^pos^) and mature DCs (CD80^pos^, CD86^pos^, CD83^pos)^	Human	BM-MSCs	In vitro	[43]
	↓ secretion of TNF and IL-12	Human	BM-MSCs	In vitro	[44]
	↓ production of osteopontin	Human	AT-MSCs	In vitro	[45]
	Induction of regulatory DCs with CD11c^pos^, MHCII^high^, CD80^low^, CD86^low^, DEC205^low^ phenotype	Mice	ND	In vitro, In vivo	[46]
	↓ expression of CD40, ↓ secretion of TNF	Mice	AT-MSCs	In vitro	[47]
Natural killer cells (NK cells)	↓ proliferation and cytotoxic activity (synergistic action with IDO)	Human	BM-MSCs	In vitro	[50]
B cells	↑antibody production (IgM, IgA, IgG); ↑ proliferation	Human	UC-MSCs	In vitro	[57]
	↓ production of IgE by LPS/IL-4 stimulated B cells isolated form mice with induced allergic conjunctivitis	Mice	BM-MSCs	In vivo	[58]
	↓ maturation (% CD27^pos^/CD19^pos^)	Human	AT-MSCs	In vitro	[59]
	↓ secretion of IL-10 by LPS-stimulated mice B cells	Mice	BM-MSCs	In vitro	[60]
	↑ secretion of IL-10 by human CD23^pos^CD43^pos^ B regulatory subset	Human	BM-MSCs	In vitro	[61]
T cells	↓ activation and proliferation of T cells	Human	AT-MSCs WJ-MSCs	In vitro	[70]
	↓ T cells proliferation	Human	BM-MSCs	In vitro	[72]
	↓ T cells proliferation by MSC exposed to apoptotic cells	Human	UC-MSCs	In vitro	[74]
	↓ T cells proliferation by MSC spheroids	Human	BM-MSCs	In vitro	[75]
	induction of human CD4^pos^ CD25^High^ FoxP3^pos^ T cells	Human	BM-MSCs	In vitro	[77]
	induction of IL-10 ^pos^IFN-γ^pos^CD4 ^pos^ regulatory T type 1 (T(R)1)-like cells	Pigs	BM-MSCs	In vitro	[79]
	↓ inflammation by increasing FOXP3 Tregs in mice model of colitis	Mice	*AT*-MSCs	In vivo	[81]
	↑ percentage of Treg in spleen and kidney in mice with ischemia-reperfusion acute kidney injury	Mice	BM-MSCs	In vitro	[82]
	↑ secretion of CCL12 and CCL5 (↑chemoattraction of T cells towards MSCs), ↓ proliferation of Tc, ↑ production of Treg	Rats	*ND*	In vitro	[83]
	inhibition of follicular helper-like T cells	Human	UC-MSCs	In vitro	[85]
	↓ Th17 differentiation	Mice	BM-MSCs	In vitro	[86]
	↓ Th17 differentiation; ↓production of IL-17, IL-22, IFN-gamma, and TNF-alpha by fully differentiated Th17	Human	BM-MSCs	In vitro	[87]

MSC—mesenchymal stem/stromal cells, BM—bone marrow, AT—adipose tissue, US—umbilical cord, WJ—Wharton’s Jelly, Tregs—T regulatory cells, ND—not determined.

## 3. The Role of PGE2 on Other Physiological Properties of MSCs

### 3.1. Supporting Hematopoiesis

It has been shown that eicosanoids, including PGE2, are involved in the physiological regulation of hematopoiesis by enhancing HSCs homing, survival and self-renewal [90]. DeGowin et al. showed that BM-MSCs, through PGE2 secretion, could regulate the growth of erythroid colonies. The authors demonstrated that MSC-derived PGE2 affected the response of erythroid colonies to erythropoietin in a dose-dependent manner [91]. Singh et al. investigated the effect of exogenous PGE2 on hematopoietic regeneration and survival after total body irradiation (TBI) in mice. Presented data suggest that PGE2 acting through the EP4 receptor enhances hematopoietic niche regeneration after TBI by increasing the survival and expansion of BM-MSCs, endothelial cells, and osteoblasts and their precursors [92]. These data, although sparse, indicate that PGE2 is one of the factors determining the basic physiological function of MSCs, which is to support the process of hematopoiesis.

### 3.2. Proliferation

One of the core features of MSCs is their ability to proliferate and self-renew. There are studies suggesting that PGE2 may also be responsible for regulating and maintaining these functions. Lee et al. describes the role of PGE2 autocrine signaling in human MSCs from adipose tissue (hAT-MSCs) and umbilical cord blood (hUCB-MSCs) [93]. The authors showed that PGE2 affected the proliferation of MSCs by using the COX-2 pathway chemical inhibitors or siRNA. Restraint of PGE2 production with COX-2 or microsomal prostaglandin E synthase-1 (mPGES-1) inhibitors resulted in a consistent reduction in both hAT-MSCs and hUCB-MSCs proliferation. The addition of PGE2 to cell culture restored proliferation capacity in a dose-dependent manner. The authors concluded that PGE2 produced by MSCs contributes to the maintenance of self-renewal capacity through EP2 in an autocrine manner, and PGE2 secretion is down-regulated by cell-to-cell contact, attenuating its immunomodulatory potency [93]. PGE2 acting through the receptor EP2 leads to an increase in intracellular level of cyclic adenosine-3′, 5′-monophosphate (cAMP). There are two families of cAMP effectors containing an evolutionarily conserved binding domain: protein kinase A (PKA) and a directly cAMP-activated exchange protein (Epac) [94]. Epac binds to cAMP leading to activation of small GTPases from the Ras protein family: Rap1 and Rap2. PKA can also phosphorylate Rap1 at its C-terminus. However, this pathway is not necessary for cAMP-dependent Rap1 activation [95]. According to Jang et al., activation of both PKA and Epac is required in the PGE2-induced increase in hUCB-MSC proliferation. The authors noted an accelerated activity of Rap1 leading to phosphorylation of protein kinase B (Akt) using the Epac agonist but not the PKA, indicating that activation took place on the cAMP/Epac pathway. Moreover, the results showed that the joint Epac/Rap1 and PKA signaling led to the phosphorylation of the β-isoform of glycogen synthase 3 (GSK-3b), which in turn resulted in the translocation of β-catenin into the cell nucleus [96]. The same work also showed that PGE2-mediated β-catenin promotes G1 phase progression and G1/S transition by increasing c-Myc and vascular endothelial growth factor (VEGF) expression, as confirmed in other MSC studies [97,98].

Kleiveland et al. noted that of the two PKA isoforms, type II presumably mediates PGE2-induced phosphorylation of GSK-3b. On the other hand, it has also been observed that activation of type I PKA leads to cell cycle arrest in the G0/G1 phase. These data suggest that the two PKA isoforms have different functions in human BM-MSCs, playing a key role in controlling proliferation. It seems that the activation of particular types of PKA in MSCs depends on the concentration of PGE2—concentration that is too high or too low will lead to activation of PKA type I and thus decrease the rate of proliferation and arrest of the cell cycle [99]. The reason for this phenomenon can be the diverse locations of PKA subtypes. Type I PKA is soluble and preferentially located in the cytosol [100], while type II PKA targets subcellular structures via kinase A anchor proteins (AKAP) [101]. A similar effect on the proliferation of MSCs as in the study of Kleiveland et al. was obtained on equine BM-MSCs. Cells treated with lower concentrations of NSAIDs (0.1–1 μM for celecoxib and meloxicam and 10–50 μM for flunixin) showed a positive effect on proliferation, while at higher concentrations (10–200 μM for celecoxib and meloxicam and 100–1000 μM for flunixin and phenylbutazone), proliferation was inhibited [102]. Another study in which aspirin was investigated also confirmed the role of PGE2 in regulating the MSCs’ proliferation rate. Growth rate inhibition and a decrease in DNA synthesis were observed in rat BM-MSCs cultured in the presence of the drug, while cell cycle analysis showed no changes in the proportion of cells at different stages of the cell cycle. Additionally, Western blot analysis showed an increased level of phosphorylated β-catenin protein in MSCs after aspirin treatment, which confirms the reports described above [103]. 

### 3.3. Migration

From a clinical point of view, a significant feature of MSCs is their ability to migrate towards the inflamed area, which allows these cells to act locally at the site of injury [104]. For example, it was shown in vitro that in the presence of damaged lung cells, murine BM-MSCs displayed a greater potential to migrate and proliferate than in the presence of healthy cells [105]. These abilities have been confirmed in many other in vitro studies showing increased migration of MSC in response to various chemotactic factors, such as stromal cell-derived factor-1 (SDF-1), platelet-derived growth factor AB (PDGF-AB), insulin-like growth factor-1 (IGF-1), epidermal growth factor (EGF), hepatocyte growth factor (HGF), interleukin (IL) -1b, TNF, CC chemokine ligand 5 (CCL5/RANTES) and macrophage-derived chemokine (MDC/CCL22) [106,107,108].

There are many reports demonstrating that PGE2 may play an important role in the migration capacity of various cell types, including, but not limited to, dendritic cells (DCs) [109,110,111], HSCs [92,112], macrophages [113], and cancer cells [114,115]. It was reported that PGE2 also plays a role in the migration capacity of MSCs. Yun and colleagues suggested that PGE2 could increase MSC migration capacity. The authors demonstrated that PGE2 partially stimulates human MSCs migration and proliferation by interacting with Pfn-1 and F-actin via EP2 receptor-dependent β-arrestin-1/JNK signaling pathways [116]. In turn, the results of Lu et al. indicated the role of PGE2 as a signal for EP2-expressing MSCs to navigate them to reach damaged tissues. The authors also proved that activation of FAK and ERK1/2 is necessary for PGE2-induced human BM-MSC migration [117]. At the same time, in our previous in vitro study, inhibition of PGE2 production through ibuprofen treatment used in therapeutic concentrations showed no changes in human BM-MSCs migration, both in the overall mobility (scratch test) and direct migration (transwell assay) tests [118]. These data may indicate that PGE2 is one of many potential stimulators of the migratory capacity of MSCs, but in the case of reduced PGE2 concentration, the presence of other factors is sufficient to maintain the basic mobility of this population [118].

### 3.4. Differentiation

Another essential feature of MSCs used in cellular therapy is their multi-lineage differentiation capacity. At least in some applications, i.e., bone defects, transplanted MSCs are expected to differentiate into target tissue in response to environmental signals. The role of COX-2/PGE2 in these processes seems to be ambiguous. 

Many in vivo and in vitro studies show the critical role of COX-2/PGE2 in bone formation [119,120,121] and the negative influence of NSAIDs on this process [122,123]. It has been shown that PGE2 can stimulate new bone formation, increase bone mass and regulate the expression of proteins related to osteogenesis such as bone morphogenetic protein 2/7 (BMP-2/BMP-7) [119,124,125]. Animal studies showed that injection of PGE2 stimulates osteogenesis in rats [120,121], whereas mice with *COX-2* knockout have a decreased level of bone density [126]. 

The role of MSCs in the regulation of bone formation through the COX-2/PGE2 pathway has also been investigated. A study by Zhang et al. on bone marrow MSCs obtained from *COX2*(-/-) mice displayed a defect in osteogenesis which was ultimately rescued after adding of PGE2 to the cell cultures. Additionally, researchers proved the role of COX-2 in the induction of *Cbfa1* and *Osx*—two genes necessary for bone formation [127]. Another group determined that in human BM-MSCs, the expression of BMP-2, a potent osteoinductive cytokine, is stimulated by COX-2/PGE2 with the EP4 receptor involvement [125]. Keila et al. tested the hypothesis that PGE2 promotes bone marrow MSCs osteogenic differentiation both in vivo and in vitro [128]. Their results show an approximately 4-fold increase in the number of mineralized nodules in ex vivo BM-MSCs cultures from rats treated for two weeks with PGE2 (5 mg/kg/day). Moreover, the alkaline phosphatase activity was also significantly higher (by 30–40%) compared to the control without PGE2 treatment. In vitro PGE2 treatment reversed the aging-related decline in osteogenic differentiation (nodule formation) of rat bone marrow MSCs [128]. Another study investigated the effect of selective (celecoxib) and non-selective (indomethacin) COX-2 inhibitors on the osteogenesis potential of murine BM-MSCs from the bone marrow. The analysis of the mRNA expression level of *Bglap2* (osteocalcin) and *Col1a1* (collagen Iα) did not unambiguously indicate the effect of the tested NSAIDs [129]. 

In turn, another study on mice showed that COX-2 expression in bone fractures in older animals (52–56 weeks) was statistically significantly lower compared with young (7–9 weeks) animals by 75% on day 5 and by 65% on day 7 of the experiment. It was associated with a clinical effect—a reduced rate of chondrogenesis and bone formation and remodeling. The highest level of COX-2 expression was recorded on the fifth day, which correlated with chondrogenic differentiation of mesenchymal cells. Local administration of COX-2/EP4 agonists compensated the losses in signal transmission and restored the level of bone repair in aged animals to that observed in young individuals. These results suggest that the expression of COX-2 during the early inflammatory phase in bone lesions is an essential factor regulating other repair processes, including chondrogenesis [130].

All results presented above were obtained on rodent models and they all coherently indicate that PGE2 enhance osteogenesis. Pountos and colleagues analyzed human MSCs and came to slightly different conclusions. They investigated whether disorders in bone formation associated with the NSAIDs administration were related to the inhibition of bone marrow MSCs osteogenic and chondrogenic differentiation [131]. Their results showed no effect on the human MSCs osteogenesis; however, chondrogenic differentiation was disturbed. They reported that the content of sulfated glycosaminoglycans (sGAG) after chondrogenic differentiation was reduced by approximately 50% in diclofenac- and ketorolac-treated MSCs in comparison to control. More specific inhibitors of COX-2 such as parecoxib and meloxicam decreased sGAG formation to a lesser extent (approximately 25%) [131]. These findings suggest that NSAIDs may disturb the proper bone formation processes by inhibiting MSC chondrogenic differentiation, a crucial intermediate phase in normal endochondral bone formation. Another group presented similar conclusions [132]. They noticed that naproxen downregulates mineral deposition in human MSCs cultured in an osteogenic differentiation medium and affects gene expression, including *COL10A1* (collagen type X), which is associated with endochondral ossification. A decrease in the expression of the alkaline phosphatase gene and its enzymatic activity was also described, which suggests that naproxen could negatively affect the potential MSC-mediated repair of subchondral bone. Other studies have confirmed that naproxen increases collagen type X expression at the transcriptional and translational levels in human MSCs [133,134]. Yoon et al. investigated the effect of COX-2 inhibitors (celecoxib and naproxen) on the osteogenic potential of human bone marrow MSCs. A decrease in the calcium deposition and the activity of alkaline phosphatase was observed in the tested cells in comparison to control. In addition, the expression of genes related to osteogenesis, *RUNX2*, *DLX5*, and *BGLAP* (osteocalcin) was decreased after treatment with NSAIDs in comparison to control in a dose-dependent manner [135]. Studies on the influence of PGE2 on MSC differentiation suggest the existence of interspecies differences in regulating these processes. This indicates the importance of verifying the results of research conducted on rodents on human material.

## 4. Cancerogenesis

Despite the well-known potentially therapeutic properties of MSCs and numerous scientific reports on the possible applications of these cells in the treatment of various diseases, this population also has a dark side. It has been proven that MSCs play a significant role in the formation, growth, progression and metastasis of neoplastic cells [136,137]. MSCs are recruited to the site of tumor formation, where they co-create and modify a microenvironment conducive to neoplasia [138]. Moreover, in 2004, a paper published in *Science* indicated that bone marrow-derived MSCs could undergo direct transformation into cancer cells in the environment of chronic stomach inflammation induced by Helicobacter pylori, a known carcinogen [139]. 

PGE2 is a molecule contributing to MSC-assisted tumor development [140]. Increased expressions of COX-2 and PGE are found in many neoplastic tissues favoring their formation and progression as well as angiogenesis processes [141,142]. An increase in the COX-2/PGE2 axis activity was noted not only in MSCs but also in other populations inhabiting the neoplastic niche, such as myeloid-derived suppressor cells (MDSCs) and tumor-associated macrophages [143]. 

The work of Li et al. provided a potential explanation of PGE2-mediated role of MSCs in carcinogenesis [144]. According to the presented results, BM-MSCs in response to interleukin-1 (IL-1) produced by carcinoma cells secrete increased amounts of PGE2. The resulting PGE2, together with other cytokines also secreted by MSCs, contributes to the transformation of neighboring cancer cells into a stem cell-like state. Another study showed that the hypoxia prevailing in the tumor microenvironment increases the expression of COX-2 in MSCs, thus stimulating the synthesis of PGE2. Through the EP4 receptor, PGE2 activates the Yes-associated protein (YAP), leading to increased lipogenesis in hepatocellular carcinoma. Intensive lipogenesis processes deliver building elements to the tumor, promoting rapid cell proliferation [145]. Another mechanism of action of MSCs in supporting cancer development was proposed by Naderi et al. Their research was based on the fact that PGE2 produced by human umbilical MSCs can affect the cyclic adenosine monophosphate (cAMP) signaling pathway, which, in turn, by activating protein kinase A (PKA), inhibits p53 accumulation caused by DNA damage, as well as protect cells from apoptotic death. Their results confirmed that bone marrow-derived resident MSCs support the development of B cell precursor acute lymphoblastic leukemia by influencing the PGE (2)-cAMP-PKA signaling pathway [146]. Another role for the COX-2/PGE2 pathway in tumor-associated MSCs was demonstrated by the team of Martinet et al. [147]. In their studies, human BM-MSCs have been shown to be potent suppressors of human T_γδ_ cells proliferation, cytokine production and in vitro cytolytic responses. Increased levels of COX-2 and PGE2 expression were stimulated by IFN-γ and TNF secreted by lymphocytes [147].

Two different potential roles have been described for MSCs in the development of triple-negative breast cancer (TNBC) with BRCA1 gene mutation (aka IRIS, for In-frame Reading of BRCA1 Intron 11 Splice variant), which is the most aggressive breast cancer subtype. First, tumor cells have been observed to secrete high levels of IL-6, which upregulate the expression of the IL-6 receptor (IL-6R) in MSCs while enhancing their proliferation and migration towards the tumor both in vitro and in vivo. The in vitro co-culture with TNBC-IRIS cells showed a shift in the profile of MSCs towards pro-aggressiveness cancer-associated fibroblasts (CAFs) characterized by a high level of PGE2 secretion. The results were confirmed in vivo, where co-administration of both cell populations promoted the formation of aggressive mammary tumors and reduced overall survival of animals compared to the group administered with a single TNBC-IRIS cell population. On the other hand, it was also suggested that naïve MSCs can contribute to the death of tumor cells with decreased expression of IRIS, but the mechanism of action behind this activity has not been elucidated [148]. 

## 5. Conclusions

Overall, COX-2/PGE2 plays an important role in maintaining the basic vital functions of MSCs such as proliferation, migration and differentiation capacities. The results published so far generally indicate that, according to the physiological principle of redundancy, PGE2 is a co-responsible rather than a critical factor for the mentioned processes. In regard to the modulating the external environment, published data indicate that MSC-derived PGE2 positively influences the creation of unique niches conducive to processes such as hematopoiesis and carcinogenesis. So far, in regard to the effect of PGE2 on the immunomodulation of human MSCs, the most consistent results seem to relate to the inhibitory effect on the early differentiation of dendritic cells and on the proliferation of activated T cells. The data indicating the impact of MSC-derived PGE2 on the differentiation of macrophages in majority come from murine studies and should be further confirmed in human material. Data on the role of PGE2 in interactions between MSCs and B cells seem insufficient to draw clear conclusions. The need for further research in this area is indicated. Generally, studies conducted in the field of immunomodulation reveal the difficulty of unambiguous determination of the role of a single factor in highly dynamic co-culture systems, especially with the participation of responsive cell types that are in a mutual dependency. Moreover, the role of COX-2/PGE2 pathways is often verified with the use COX inhibitors, which are known to have a wide panel of specificity to certain COX isoforms. The use of different COX inhibitors at different doses and different experimental setups makes it very difficult or even impossible to draw common conclusions from different studies. Therefore, trends leading to a relative standardization of research protocols would be advisable.

## Figures and Tables

**Figure 1 biomedicines-11-00445-f001:**
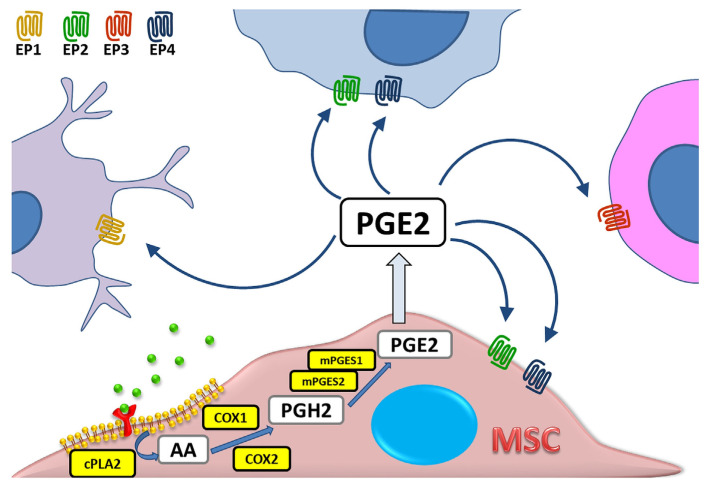
Schematic presentation of prostaglandin E2 (PGE2) biosynthesis in a mesenchymal stromal cell (MSC). AA—arachidonic acid, PGH2—Prostaglandin H2, cPLA—cytosolic phospholipase A2, COX—cyclooxygenase (alias names: PTGS—Prostaglandin endoperoxide synthase and PGHS—prostaglandin G/H synthase), mPGES—microsomal prostaglandin E synthases. PGE2 exerts its effect via one of four receptors (EP1–EP4).

**Figure 2 biomedicines-11-00445-f002:**
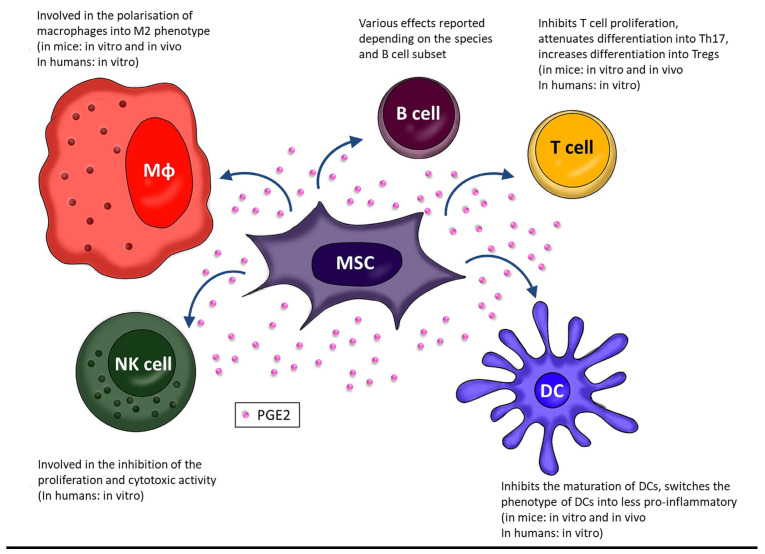
A scheme summarizing current data on the immunomodulatory effects of MSCs on specific immune cell types exerted via the COX-2/PGE2 pathway.

## Data Availability

Not applicable.

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
