# Peer review of "The Role of COX-2 and PGE2 in the Regulation of Immunomodulation and Other Functions of Mesenchymal Stromal Cells"

_biomedicines, 2023, doi:10.3390/biomedicines11020445_

Round 1
Reviewer 1 Report
The authors and the journal are to be congratulated on this wellwritten and most informative review regarding the studies performed regarding the role of Cox-2 and PGE2 regulation regarding variouse functions of MSCs.
I only miss one effect by PGE2 on MSCs,the inhibitory effect on alloreactivity in vitro by fetal membrane derived so called decidua stromal cells(DSCs).
Different sources of MSCs have different properties.MSCs from bone marrow (BM)are those most widely studied.However,all tissues contain MSCs ,although at low levels,less than 1:10000 nucleated cells.However,MSCs from different tisuues may have different properties.It will be helpful for the readers of this excellent review,to know the sources of MSCs used in the various reports.This is done for many studies.However,if the authors would add if MSCs were from bone marrow(BM),adipose (Ad),Umbilical Cord(UC),placenta tissue (PT),amnion (An),decidua stroma (DSC) etc throughout all references used should be most helpful to the readers of this excellent review.
IE BM-MSCs, Ad-MSCs,UC-MSCs etc.should be clarified throughout all references.
Author Response
R: The authors and the journal are to be congratulated on this wellwritten and most informative review regarding the studies performed regarding the role of Cox-2 and PGE2 regulation regarding variouse functions of MSCs.
A: Thank you very much for appreciating our work
R: I only miss one effect by PGE2 on MSCs,the inhibitory effect on alloreactivity in vitro by fetal membrane derived so called decidua stromal cells(DSCs).
A: The fragment regarding the DSCs-derived PGE2 on immune cells has been added to the manuscript with the appropriate reference.
R: Different sources of MSCs have different properties.MSCs from bone marrow (BM)are those most widely studied.However,all tissues contain MSCs ,although at low levels,less than 1:10000 nucleated cells.However,MSCs from different tisuues may have different properties.It will be helpful for the readers of this excellent review,to know the sources of MSCs used in the various reports.This is done for many studies.However,if the authors would add if MSCs were from bone marrow(BM),adipose (Ad),Umbilical Cord(UC),placenta tissue (PT),amnion (An),decidua stroma (DSC) etc throughout all references used should be most helpful to the readers of this excellent review.
IE BM-MSCs, Ad-MSCs,UC-MSCs etc.should be clarified throughout all references.
A: Thank you for this remark. Indeed, it is worth indicating the source of the MSCs. The manuscript has been corrected – now, when citing original papers, it is specified which type of MSC was used in the study. Moreover, we have added a table showing the mechanisms of action of MSCs on immune cells via PGE2 and we also provided the source of MSCs in there.
We resubmit the manuscript. The word version is provided with tracked changes which will make it easy to see the corrections we have made. In turn, the attached PDF is in the version after accepting the changes to make it easier to read.
Reviewer 2 Report
Dear Authors,
The manuscript entitled "The role of COX-2 and PGE2 in the regulation of immunomodulation and other functions of mesenchymal stromal cells" is a very well written review. Only the following revisions must be performed in order the manuscript to be further improved.
1) Please include a figure summarizing the properties of PGE2 and its relationship with the cells of the immune system.
2) Please also include a table with the properties of PGE2 to the main manuscript.
3) Please add information regarding the biology of MSCs and their immunoregulatory properties.
4) Please add the following references regarding the biology of MSCs to your manuscript
a) https://doi.org/10.37349/ei.2021.00010
b) doi: 10.4331/wjbc.v13.i2.47.
Author Response
The manuscript entitled "The role of COX-2 and PGE2 in the regulation of immunomodulation and other functions of mesenchymal stromal cells" is a very well written review. Only the following revisions must be performed in order the manuscript to be further improved.
A: Thank you very much for reviewing our manuscript and providing valuable comments and suggestions. We believe that the revised manuscript is of improved quality.
1) Please include a figure summarizing the properties of PGE2 and its relationship with the cells of the immune system.
A: We have added a figure summarizing the MSC-derived PGE2 on different types of immune cells. Moreover, we have included one more figure presenting schematically the PGE2 biosynthesis to help the reader visualize this process, which is central for the article.
2) Please also include a table with the properties of PGE2 to the main manuscript.
A: According to the suggestion, we included a table presenting identified mechanisms by which MSCs exert the effect on activity of immune cells via COX-2/PGE2 pathway.
3) Please add information regarding the biology of MSCs and their immunoregulatory properties.
A: Additional information regarding basic biological features of MSCs has been added to the manuscript in the “Introduction” section. Since the description of the entire panel of identified immunomodulatory mechanisms associated with MSCs is beyond the scope of our review article, we have only listed other molecules involved in MSC immunomodulation and cited two recent review articles to allow the readers easy access to extended information on this topic.
4) Please add the following references regarding the biology of MSCs to your manuscript
- a) https://doi.org/10.37349/ei.2021.00010
- b) doi: 10.4331/wjbc.v13.i2.47.
A: The reference (a) has been added to the manuscript. We have decided not to add the reference (b) as it does not specifically refer to the activity of MSC-derived PGE2.
We resubmit the manuscript. The word version is provided with tracked changes which will make it easy to see the corrections we have made. In turn, the attached PDF is in the version after accepting the changes to make it easier to read.
Round 2
Reviewer 2 Report
Dear Authors,
You have respond correctly to all my comments. Good work!!